# High Inclusiveness and Accuracy Motion Blur Real-Time Gesture Recognition Based on YOLOv4 Model Combined Attention Mechanism and DeblurGanv2

Hongchao Zhuang [1,*], Yilu Xia [1], Ning Wang [2] and Lei Dong [1]

1   School of Mechanical Engineering, Tianjin University of Technology and Education, Tianjin 300222, China; xiayilu97@126.com (Y.X.); donglei_hit@163.com (L.D.)
2   School of Information Technology Engineering, Tianjin University of Technology and Education, Tianjin 300222, China; wangning811108@163.com
*   Correspondence: zhuanghongchao_hit@163.com; Tel.: +86-022-8818-1083

**Abstract:** The combination of gesture recognition and aerospace exploration robots can realize the efficient non-contact control of the robots. In the harsh aerospace environment, the captured gesture images are usually blurred and damaged inevitably. The motion blurred images not only cause part of the transmitted information to be lost, but also affect the effect of neural network training in the later stage. To improve the speed and accuracy of motion blurred gestures recognition, the algorithm of YOLOv4 (You Only Look Once, vision 4) is studied from the two aspects of motion blurred image processing and model optimization. The DeblurGanv2 is employed to remove the motion blur of the gestures in YOLOv4 network input pictures. In terms of model structure, the K-means++ algorithm is used to cluster the priori boxes for obtaining the more appropriate size parameters of the priori boxes. The CBAM attention mechanism and SPP (spatial pyramid pooling layer) structure are added to YOLOv4 model to improve the efficiency of network learning. The dataset for network training is designed for the human–computer interaction in the aerospace space. To reduce the redundant features of the captured images and enhance the effect of model training, the Wiener filter and bilateral filter are superimposed on the blurred images in the dataset to simply remove the motion blur. The augmentation of the model is executed by imitating different environments. A YOLOv4-gesture model is built, which collaborates with K-means++ algorithm, the CBAM and SPP mechanism. A DeblurGanv2 model is built to process the input images of the YOLOv4 target recognition. The YOLOv4-motion-blur-gesture model is composed of the YOLOv4-gesture and the DeblurGanv2. The augmented and enhanced gesture data set is used to simulate the model training. The experimental results demonstrate that the YOLOv4-motion-blur-gesture model has relatively better performance. The proposed model has the high inclusiveness and accuracy recognition effect in the real-time interaction of motion blur gestures, it improves the network training speed by 30%, the target detection accuracy by 10%, and the value of mAP by about 10%. The constructed YOLOv4-motion-blur-gesture model has a stable performance. It can not only meet the real-time human–computer interaction in aerospace space under real-time complex conditions, but also can be applied to other application environments under complex backgrounds requiring real-time detection.

**Keywords:** gesture recognition; YOLOv4; CBAM; deep learning; image blur restoration

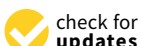



## 1. Introduction

With the research and development of robotics, machine vision has been able to replace the human eyes for certain target recognition and judgment. The visual recognition is very important for robots, especially for the electrically driven large load-ratio multi-legged robots in planetary exploration [1,2]. The intelligent target recognition of the robots can be realized by applying deep learning in the robot vision task. Gesture recognition is widely used in the interaction with robots in aerospace space with complex and harsh conditions.

By making different command gestures to the robots, the command transmission can be completed. Before the deep learning is widely adopted, the feature representation in the traditional target detection is generally based on the manual design of feature detectors [3,4], such as SIFT [5], HOG [6], and so on. These traditional target detections are based on the content of a picture, which is difficult to design, low in portability, and low in recognition accuracy. Taking the face detection in the picture as an example, the hand-designed features usually need to repeatedly adjust the relevant parameters to fully detect the target in an ideal environment. Early gesture recognition tasks require the users to wear data gloves to perceive and complete the recognition. For example, the Mistry team designed a gesture recognition interface WUW [7] based on the wearable video devices. The interface uses a wearable camera to capture the gesture images. The gesture images can be converted into the input signal operation commands. Although the recognition efficiency of that method is high, it is inconvenient to use. The equipment is heavy and cumbersome to wear. With the continuous development and optimization of wearable devices, researchers also make them into the shape of wristbands and finger rings. However, their recognition and perception are still based on the transmission of wires. There are still certain difficulties in popularization and use of wearable devices [8,9]. For a more convenient and natural user experience, the gesture recognition based on the machine vision has developed rapidly. To complete non-contact gesture recognition, the study of gestures is performed through the computer by researchers. The gesture images are collected by camera. By performing the gesture segmentation, gesture tracking, and gesture feature extraction, the information can be transmitted by the computer.

Based on a screen-printed conformal electrode array, Moin et al. put forward a wearable surface electromyography biosensing system [10]. That system has sensor adaptive learning capabilities which can be applied to real-time gesture classification. Song et al. designed a virtual reality interactive glove by using an actuator, which is very light and easy for users to wear [11]. Lee et al. presented a skin-patchable magneto-interactive electroluminescent display, which is capable of sensing, visualizing, and storing magnetic field information to realize the 3D motion tracking [12]. Mantecón et al. proposed a hand gesture recognition system by using near-infrared imagery. The hand gesture characterizes can be recognized by the system directly [13]. Esteva et al. used the CNN in many recognition and classification applications to achieve the machine learning [14]. Wang et al. performed the method research of human gesture recognition by integrating the visual data and somatosensory data from the skin-like stretchable strain sensors made from single-walled carbon nano-tubes [15]. That method is neither limited by the quality of the sensor data nor the incompatibility of the datasets. It can maintain the recognition accuracy under the nonideal conditions of the high image noise, underexposure, or overexposure. Shinde et al. utilized YOLO to complete the recognition and positioning of human actions, which can accurately identify and locate the group frames or even single frames of human movements in the video [16]. Yu et al. proposed a face mask recognition and standard wear detection algorithm based on the improved YOLO-v4 [17]. The backbone network of the model is improved. The adaptive image scaling algorithm is proposed. Additionally, the face mask detection data set is made according to the standard wearing of masks. The model proposed can improve the face mask recognition effectively. Roy et al. proposed a deep learning enabled object detection model for multi-class plant disease to detect different apple plant diseases under complex orchard scenarios [18].

Noncontact gesture recognition has a good development prospect in the field of human–computer interaction. Due to the complexity and variety of gestures, there are great differences and uncertainties making gesture recognition more difficult. Noncontact gesture recognition can be performed by segmenting and recognizing the color area or the contour edge of the gesture. However, when the background and hand colors are similar or the background is complex, the gestures based on machine vision cannot extract the target features well. The recognition effect is not good. The image captured by the camera is blurred when the human hand moves. The contour of the hand is not clear under

the that situation. The feature extraction and region segmentation are difficult [19]. Due to radiation, space station fluctuations, and other factors, the gestures captured by the machine are usually blurred. The bur gesture images are not conducive to the recognition and judgment of the robots and will cause command transmission delay or error. In the abominable aerospace operations, a gesture recognition system with high recognition efficiency and high tolerance can bring higher work efficiency.

To solve the above problems and facilitate the elaboration of this article, four different gestures are utilized in the research of the actual human–computer interaction environment. This article is divided into five parts. In Section 2, the K-means++ clustering algorithm is used to cluster the labeled dataset for obtaining the appropriate prior frame sizes in the target detection. In addition, the network structure is modified, such as adding the CBAM and SPP modules to improve the effect of network feature extraction and network learning attention. To ameliorate the model, the reasonable hyperparameters are chosen to adjust. The motion blur is caused by relative motion between the hand and the camera. The DeblurGanv2 is employed to remove the motion blur of gestures in the images. In Section 3, the dataset for network training is designed for the human–computer interaction in the aerospace. The expansion of the dataset is realized by simulating different environments. To reduce the redundant features of the captured images and enhance the effect of model training, the Wiener filter and bilateral filter are superimposed on the blurred images in the dataset to simply remove the motion blur. In Section 4, the built model algorithm is trained and tested on the processed gesture image through the Tensorflow framework. The ROC, PR, mAP, and other evaluation indicators are employed to verify the reasonableness of the YOLOv4-hand model. In the final section, the conclusions are presented. The YOLOv4-motion-blur-gesture model proposed in this article cannot only be used in the field of human–machine interaction robots for aerospace space exploration under complex conditions, but also can be used on other occasions with complex conditions and high requirements for the model. The schematic diagram of article content structure is shown in Figure 1.

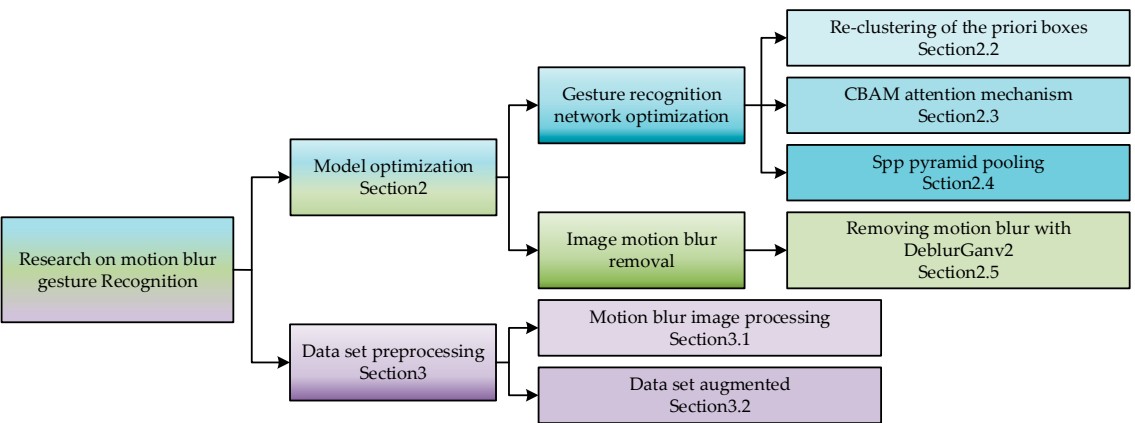

**Figure 1.** Schematic diagram of article content structure.

## 2. Algorithm Improvement

### 2.1. YOLOv4 Target Detection Algorithm

The current target detection methods can be roughly divided into two categories: one stage detection method and two stage detection method. The two stage targets detection method is represented by RCNN series models. RCNN series models first extract the proposal with the selective search method, then extract the feature with CNN, and finally train the classifier with SVM to complete the targets detection task. YOLO (You Only Look Once) is a proposal-free regression method [20]. YOLO removes the region proposal module, can directly predict whether each grid contains targets and the probability of including targets from the feature map. YOLO extracts the proposal and targets recognition

by sharing convolution features. The region proposal is constrained by using the grid. A faster training and testing speed can be obtained by avoiding repeatedly extracting proposals in some areas. On the premise that the ground truth is known in the training process, the regression equation from characteristic map to parameters such as coordinates and confidence can be established by YOLO. The model fitting curve can finally be obtained by learning the parameters of the regression equation through a lot of training. The annotation, confidence, and category probability calculation of the end-to-end bounding boxes of the targets can be completed only by inputting the picture once [21]. YOLO has high recognition efficiency and is widely used in real-time detection. The predicted value of YOLO can be represented by five elements $(x, y, w, h, C)$. Where $(x, y)$ is the center coordinate position of the bounding boxes, $w$ and $h$ are the width and height of the bounding boxes, and $C$ is the confidence level of the bounding boxes. In YOLO's classification task, each cell must also give the conditional probability value $C$ ($C \in R^+$) [22]. The conditional class probability of each grid is multiplied in the network test by the confidence of each bounding box. The specific class confidence scores of each bounding box can be obtained. The calculation method is described as follows.

$$\Pr(Class_i|Object) * \Pr(Object) * IOU_{pred}^{truth} = \Pr(Class_i) * IOU_{pred}^{truth} \tag{1}$$

where $IOU_{pred}^{truth}$ is the ratio of the intersection and union of predictor with the actual bounding boxes in the computer detection task. $\Pr(Class_i|Object)$ represents the probability that the current cell has an object and belongs to class $i$. $\Pr(Object)$ indicates the probability of the targets containing in the grid. $\Pr(Class_i)$ represents the probability that the target in the grid is class $i$.

The conditional class probability is predicted for each bounding box. We know that YOLOv4 is obtained through a series of enhancements and improvements of YOLOv3. The main contents are presented as follows.

1.  Upgrading the original backbone feature extraction network Darknet-53 to CSPDarknet-53;
2.  Enhancing the effect of feature extraction network, using the SPP and PANet structure;
3.  Utilizing the Mosaic function to complete the data enhancement;
4.  Using CIOU as return LOSS;
5.  Using Mish function as the activation function of the network.

The YOLOv4 network structure diagram is shown in Figure 2. The YOLOv4 network structure mainly includes the backbone feature extraction network (CSPDarknet53), the spatial pyramid pooling structure (SPP), and the path aggregation network (PANet).

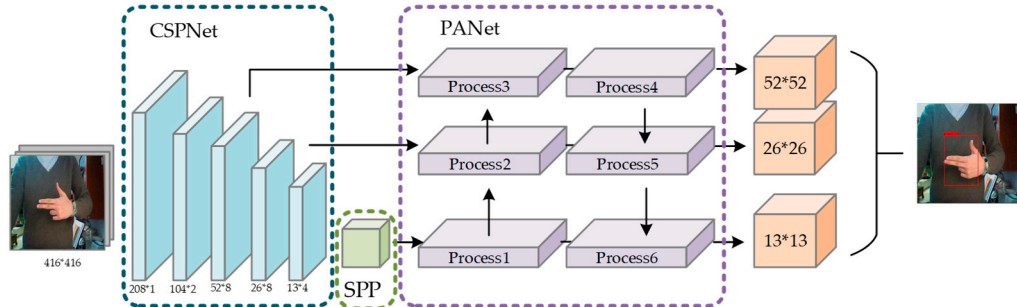

**Figure 2.** YOLOv4 network structure diagram.

### 2.2. Reclustering of Priori Boxes

The K-means clustering algorithm [23] is employed to cluster the COCO dataset in the YOLOv4 target detection algorithm. Then, 9 priori boxes of different sizes and proportions are obtained. The most common rectangular boxes in the dataset are also acquired so that the network can use the more appropriate rectangular frames to lock the targets in the subsequent target search. Consequently, the convergence of the model is accelerated.

YOLOv2 (You Only Look Once, vision 2), the K-means clustering algorithm was utilized to cluster the nine priori boxes of different sizes, which have different receptive fields and can be used to identify targets of different sizes.

The initial clustering center of the network is determined by random selection under the K-means algorithm. The initial points are more sensitive to noise and abnormal points. The unstable clustering effect is always led by the random initial points selection. The affection makes the clustering effect unstable and more sensitive to noise and abnormal points. The initial point selection of the K-means++ [24] algorithm can significantly improve the classification results and reduce the final error. The great improvement of algorithm is realized in the selection of the first clustering center. To get a more reasonable distribution, the distance between the selected initial clustering centers needs to be required as small as possible. Firstly, a sample is randomly selected from the data as the initial cluster center. The shortest distance between each sample and current existing cluster center can be calculated. Then, the probability that each sample is selected as the next cluster center can be calculated. The next cluster center can be chosen by the roulette method. After determining the next cluster center, the above steps are repeated until the appearing of $k$ cluster centers. Therefore, the selection of the initial cluster center is completed. The formula for calculating the probability of the next cluster center is defined as follows.

$$P = \frac{D(x)^2}{\sum_{x \in X} D(x)^2} \tag{2}$$

where $D(x)$ represents the shortest distance between each sample and the current existing.

In different application scenarios, the size of the target is different from the priori frame sizes which are obtained by clustering the original COCO dataset. When the difference of the target sizes is small, the original priori frame sizes with a larger threshold interval are not fully invested in the training of the target detection neural network. The result of the training network is not good. To improve the matching degree between the target gestures and the prior boxes, the K-means++ algorithm is employed to regenerate the priori boxes in the existing dataset. The priori frame size results diagram generated by K-means clustering algorithm and the K-means++ clustering algorithm are shown in Figure 3. In Figure 3, the target boxes of various scales can be divided by the K-means++ clustering algorithm clearly. The clustering effect is unstable because of the randomness of the initial point selection under the K-means clustering algorithm. The clustering results are easily disturbed by extreme value information. To obtain the more reliable clustering results, multiple clustering is usually used for comprehensive screening to get the final results. Multiple clustering calculation consumes a lot of time and there are also many uncertainties. To make improvements, the K-means++ clustering algorithm is cited in this experiment. The K-means++ clustering algorithm can make the classification results more stable by taking the target frames of various scales into account.

### 2.3. Introduction of Attention Mechanism

The size of the target pixel value can be calculated by the surrounding pixels in the convolutional neural network. However, the results calculated by local information usually lead to the loss of global information and deviation. To alleviate the information bias, not only larger convolutional filters can be used, but also deeper convolutional neural networks can be constructed. However, the relative amount of calculation climbs faster and the effect is not significant [25]. Therefore, the attention mechanism is analyzed in this article to improve the training effect of the network.

The information of interest can be located by adding the attention mechanism to the neural network. The useless information can be suppressed. The attention mechanism is mainly divided into three types: spatial attention mechanism, channel attention mechanism, and mixed attention mechanism of the two. The spatial attention mechanism is responsible for the establishment of the target positions. The channel attention mechanism is mainly responsible for the classification of the targets.

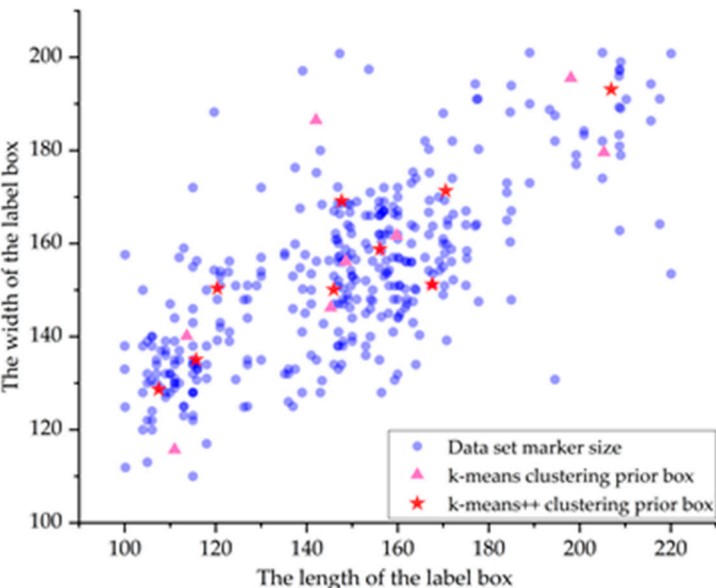

**Figure 3.** Scatter plots of prior frame sizes obtained under two different clustering algorithms.

Convolutional Block Attention Module (CBAM) is an effective light-weight feedforward convolutional neural network attention module. The structure of the CBAM attention mechanism is shown in Figure 4. The feature information of the two dimensions of space and channel are combined in the CBAM model. The CBAM model has the advantages of simple structure, small computation, and fast computation speed. It can be plugged into the network architecture as needed to improve performance. If the intermediate feature map is given, the attention map can be judged independently by the two dimensions of channel and spatial in the CBAM module. Firstly, the input features are mapped to the channel attention module. The corresponding attention mappings are output. Then, the output map is multiplied by the input characteristics and attention map, which is output through the spatial attention module. Finally, the output characteristics of the map are obtained. The mathematical expressions are written as follows.

$$F\prime = M_c(\boldsymbol{F}) \otimes \boldsymbol{F} \tag{3}$$

$$\boldsymbol{F}'' = M_s(\boldsymbol{F}\prime) \otimes \boldsymbol{F}\prime \tag{4}$$

where $\otimes$ represents element multiplication, $\boldsymbol{F}$ is the input feature map, $M_c(\boldsymbol{F})$ is the channel attention map output by the channel attention module, $M_s(\boldsymbol{F}\prime)$ is the space attention map output by the space attention module, and $\boldsymbol{F}''$ is the feature map output by the CBAM.

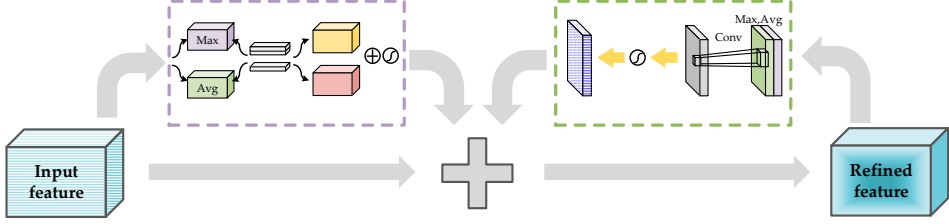

**Figure 4.** Structure diagram of CBAM attention mechanism. Where the Max represents the maximum pool processing, the Avg represents the global average pooling processing, the Conv represents the convolution layer, $\int$ represents the sigmoid activation function.

The maximum pooling and average pooling are first performed on the input image by the channel attention module. Then, the transformation results are obtained through several MLP layers. The results are respectively applied to two channels. The attention

of the channel enhancement effect is obtained by using the sigmoid function. Firstly, the maximum pooling and mean pooling are performed on the input images, respectively, under the spatial attention module. Then a convolution layer is used to learn the new feature map [26].

To obtain a more efficient attention network structure, the CBAM module is inserted among the three channels output of the CSPNet structure and the first two positions of the last five layers of each channel in the YOLOv4 network structure. The more focused learning effect of the network structure can be obtained by the above modification. The schematic diagram of adding the CBAM attention module to YOLOv4 network structure is shown in Figure 5.

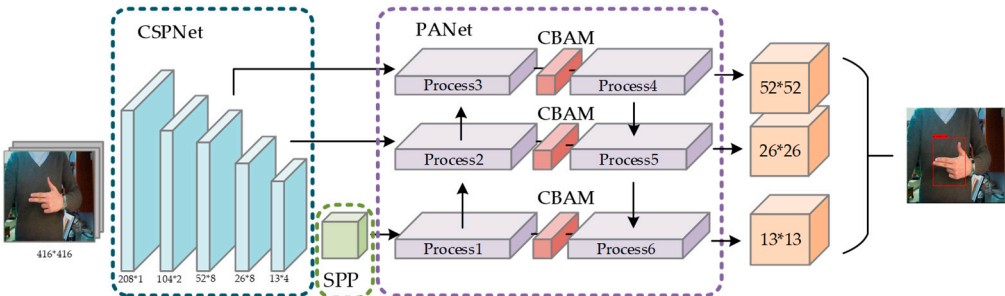

**Figure 5.** Schematic diagram of adding the CBAM attention module to the YOLOv4 network structure.

### 2.4. SPP Spatial Pooling Pyramid Structure

The SPP spatial pyramid pooling diagram is shown in Figure 6. In Figure 6, three different division methods are performed with image pooling processing by the SPP structure. The input of the SPP structure is an arbitrary size feature graph obtained by upper convolutions. Firstly, the SPP structure carries out the feature map with the sizes of $1 \times 1$, $2 \times 2$, and $4 \times 4$. Then, three different partition results are spliced. The inputs of different scales are finally normalized output to the same scale. By using different image division methods, different scales of the information can be captured. To improve the robustness, the spatial feature information of different sizes can be extracted by adding the SPP spatial pooling pyramid structure in the model [27].

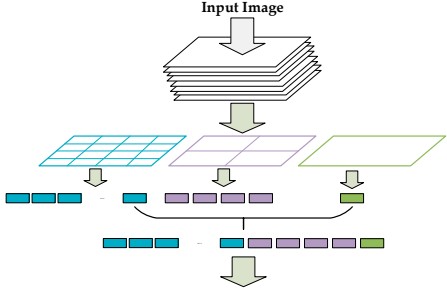

**Figure 6.** SPP spatial pyramid pooling diagram.

The YOLOv4-gesture model is improved based on the YOLOv4 model. In the YOLOv4-gesture model, the SPP spatial pyramid pooling module is added after the output of the last layer of CSPDarknet. After a convolution layer processing, the output of the last layer of CSPDarknet is processed with three different sizes of maximum pooling. The three kernel sizes are $5 \times 5$, $9 \times 9$, $13 \times 13$, respectively. The above three outputs are spliced with the original output through a connection processing. The model sensibility field can be enlarged by the maximum pooling layer while the feature map is unchanged. The three maximum pooling processes with different sizes in the SPP model can not only obtain the local receptive field of the feature map, but also obtain the near global receptive field information. The expression ability of feature map can be improved by the fusion

scale receptive field effectively. The detection performance can be improved by effectively separate important information under the SPP model. Therefore, to enhance the expression ability of feature map to receptive fields at different scales, the SPP model is added after the other two sampling ports of the backbone network. The target features of local regions and the target feature information under the global feature map can be enriched. The detection effect of the YOLOv4 model can be improved while the accuracy of target positioning and classification is increased. The schematic diagram of adding the SPP pyramid pool structure to YOLOv4 network structure is shown in Figure 7.

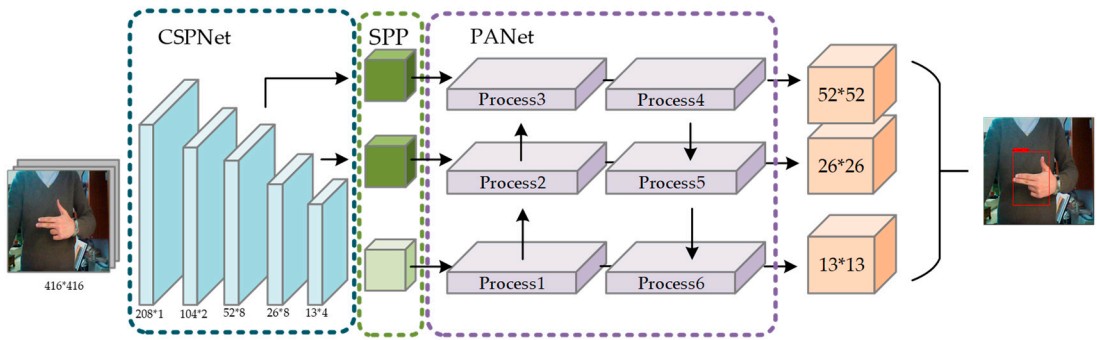

**Figure 7.** Schematic diagram of adding the SPP pyramid pool structure to the YOLOv4 network structure.

### 2.5. Motion Blur Reduction of Input Image

There is a certain relative motion between the robot camera and the hand in the process of human–computer interaction which is not easy to avoid. The decline of target recognition and judgment quality can be caused by the capture of blur gesture images. The machine recognition of unprocessed blurred images not only has a poor effect, but also often takes more time. The low tolerance of motion images makes the network unable to meet the demand of real-time human–computer interaction.

The existing image restoration tasks can be implemented by using the ResNet architecture or multi-scale input to remove image blur. However, the multi-scale method takes a long time and consumes too much memory. For the image with known motion kernel, the method of adding filter is usually used directly. Although the method of adding filter is very fast, it has no good effect on the processing of irregular blurred images. The blurred images with unknown motion kernel can be processed by the Deblurganv2 model [28]. The network structure diagram of the Deblurganv2 is shown in Figure 8. The Deblurganv2 performs feature fusion based on FPN structure. The relative discriminator is used as a discriminator. The loss is distinguished by combining global and local scales. The FPN structure includes the bottom-up and top-down paths. The bottom-up path is a convolution network for the feature extraction. The spatial resolution is down sampled in the bottom-up process. More semantic feature information is extracted and compressed. Five different scales of the final characteristic map as output are finally obtained. These features are then unsampled to a quarter of the input size and connected into a tensor. This tensor contains different levels of semantic information. An upper sampling layer and a convolution layer to restore clear images and remove artifacts are finally added in the Deblurganv2 network. At the same time, the network also introduces a jump connection from input to output in order to focus on the residual. Different image processing effects can be obtained when different backbone networks are used for training. When using a relatively complex backbone network, the image deblurring effect is better, but it takes a longer time for training. To obtain more efficient blur image processing, a lighter backbone network Mobilenet is selected for training in the real-time interactive application scenario. The Deblurganv2 not only improves the image quality after deblurring, but also can easily design a model with low computational cost. An efficient real-time processing of blurred images can be realized by using the Deblurganv2 model.

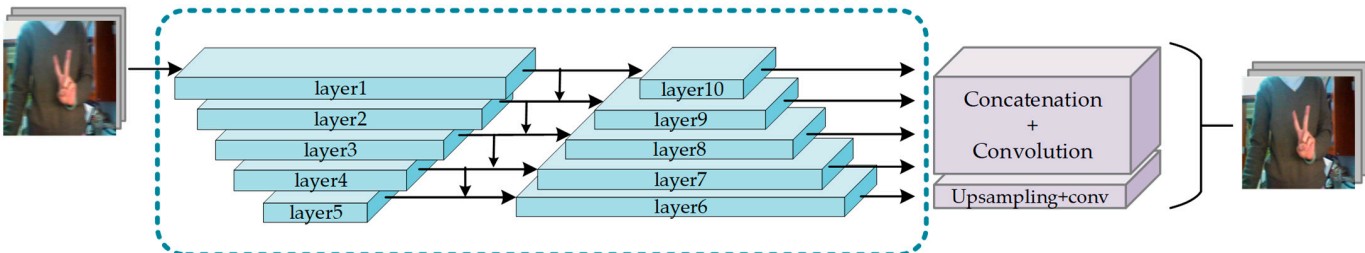

**Figure 8.** Structure diagram of the DeblurGanv2 model.

## 3. Dataset and Experimental Environment Construction

### 3.1. Dataset Establishment and Augmentation Optimization

Due to the limitation of image extraction environment, the training data of network model has high similarity and poor robustness, which cannot produce good results in the training of a neural network [29]. In the actual gesture recognition process, the difficulty of network recognition and detection can be increased by different lighting conditions, background features, hand skin color, and the distance or angle between the camera with the acquisition camera. The human–computer interaction gesture data sets in the aerospace space are difficult to obtain and limited in quantity. To make the network better adapt to the aerospace operation environment, this article enhances the original data set. The data set is processed by distorting the color gamut channel, adjusting the image scale, clipping random image, adding random illumination, adjusting the image saturation, changing the image contrast, adding filtering and noise to obtain a model with higher generalization ability. The generalization ability of the network can be increased by such adjustment. The recognition effect of different gestures can be augmented and improved by using the dataset which merged the original dataset and the enhanced images. The gesture images before and after augmentation are shown in Figure 9.

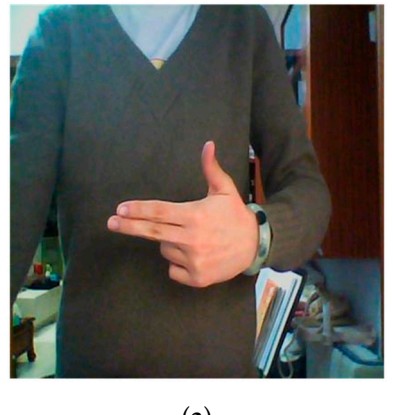
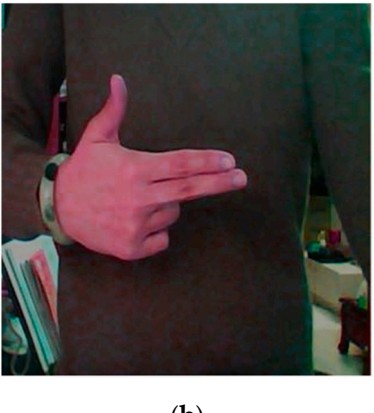

(**a**)                                          (**b**)

**Figure 9.** Gesture images before and after augmentation: (**a**) gesture image before augmentation; (**b**) gesture image after augmentation.

The neural network recognition is better when the target is complete, the background is clear, and the light is good. However, in the actual interaction process, the radiation, the fluctuation of the space station, and the movement of the hand can lead to the changes of the images collected by the robots. The information content of the image acquired will be reduced. The quality of network training and recognition is reduced by using the low information contained image. The motion blur of the collected image is caused by the movement of the hand during the actual image acquisition. The reduction of the amount of information contained in the image caused the poor network training effects. Therefore, to reduce the image motion blur, Wiener (minimum mean square error) filtering combined with bilateral filtering is selected to improve network performance.

The Wiener filtering is an adaptive minimum mean square error filter, can process pictures with noise very well [30]. The algorithm assumes that the input signal is the sum of the useful signal and the noise signal. To find the estimate of the uncontaminated images, the minimum mean square error criterion is employed. The mean square value of the difference between the filter output signal and the desired signal should be as small as possible. The image is clearer while the mean square deviation between the output signal of the filter and the desired signal is the smaller [31]. The bilateral filtering is a Gauss filtering function which is based on the spatial distribution. The values of pixels farther away do not have much effect the pixel value on the edge. The weighted average method is adopted in the bilateral filtering. The intensity of the pixel is expressed by the weighted average of the brightness values of the surrounding pixels. The motion blur images can be improved by the Wiener filtering and the bilateral filtering. Since an image is composed of many pixels, the image processing is the processing of pixels. The Wiener filtering and the bilateral filtering have different processing methods for the pixels. The effect of filtering is different with different emphasis. For the motion blurred images in the dataset, different effects can be obtained by adding a single or different filter superposition. When the appropriate parameters are selected, the processing effects of different filters on blurred images are shown in Figure 10.

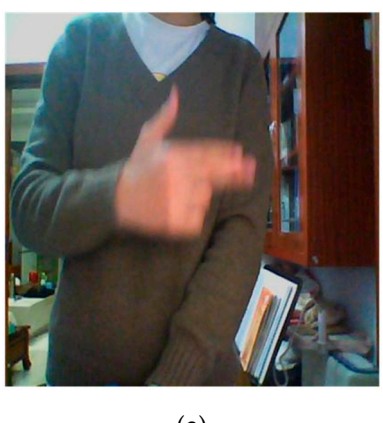 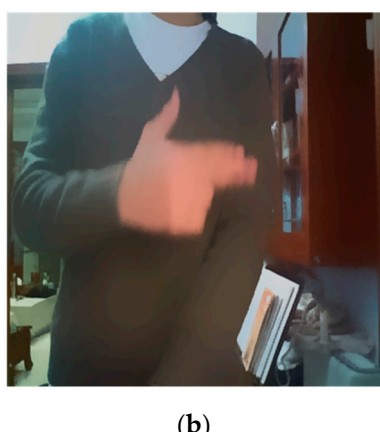 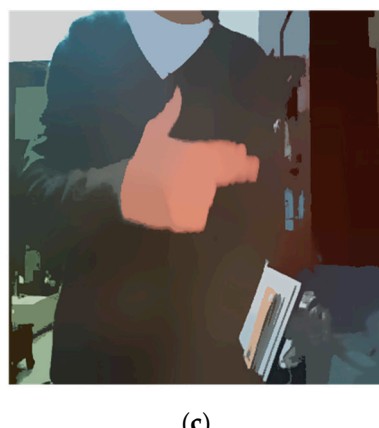

(**a**)                (**b**)                (**c**)

**Figure 10.** Motion blur image processing renderings: (**a**) original motion blur image in dataset; (**b**) motion blurred image processed by Wiener filtering; (**c**) motion blurred image processed by Wiener filtering and bilateral filtering.

In Figure 10, the motion blur image obtained for the collection is superimposed on the processing effect map after different filtering. Among them, the original motion-blurred image is given in Figure 10a. The processing effect map after superimposed Wiener filtering is shown in Figure 10b. The effect map selected after trial and error is presented in Figure 10c when the Wiener filtering and bilateral filtering are superimposed. According to Figure 10, it can be concluded that although the use of Wiener filter also has a certain effect on the processing of blurred images, the effect of using the superposition method of two filters is better.

### 3.2. Establishment of Experimental Environment

To compare the actual training and recognition effects of the models, the original model needs to be modified for experimental comparison. The performance parameters of different structural models can be obtained by training the recompiled models. Network training is divided into blur image processing module and gestures target detection module. This experiment is based on the Win10 system. The experiment hardware configuration is shown in Table 1. The experimental effect is more significant with a higher configuration. The Pychram software is used to compile the model. The Anaconda software is used to manage Python and install third-party libraries. By calling different libraries, the configuration of model training environment is configured. The model is compiled by

using a highly modular Tensorflow framework. In order to make the experimental results more comparable, each different structural model is trained in the same environment. The results were compared. The verification environment of experiment is shown in Table 2. The parameters of the training network are shown in Table 3.

**Table 1.** Experiment hardware configuration.

| System | Central Processing Unit |
|---|---|
| Win10-64bit | Intel(R) Core (TM) i7-5500U CPU @ 2.40GHz |

**Table 2.** Verification environment of the experiment.

| Model | Tensorflow | Keras | Numpy | Opencv-Python |
|---|---|---|---|---|
| DeblurGanv2 | 2.3.0 | 2.4.3 | 1.19.2 | 4.5.1.48 |
| YOLOv4 YOLOv4-gesture YOLOv4-motion-blur-gesture | 1.14.0 | 2.3.1 | 1.19.4 | 4.5.0 |

**Table 3.** Training network parameters.

| Parameter | Value |
|---|---|
| Batch size | 2 |
| Learning rate | 0.001 |
| Init_epoch | 0 |
| Freeze_epoch | 50 |
| Unfreeze_epoch | 100 |
| Fixed image size | 416*416 |

Based on VOC format datasets, the training of the neural network is executed. The verification datasets are static gesture images collected by the rear camera of the mobile phone in common use scenarios. Four different gestures are defined, they are forward, stop, victory, and holding. Each gesture has 100 pictures. The picture size is 1080*1438. Each gesture image set accounts for no less than 50% of the blurred images. The three gesture original datasets total 400. After data enhancement processing, the final dataset number is 200 images of each gesture. The final dataset is 800 in total. In general, 80% of the dataset is used to implement the network training. In total, 20% of the dataset is employed to verify the effect of network training. The annotation of the dataset is completed by using Labelimg software. The annotation diagram of the dataset is shown in Figure 11. The .xml file is generated by selecting the target gesture in the picture and entering the gesture category. After all the datasets are marked, the .xml file is converted into a .txt file, which contains the image path, target category, and location information of the target gesture. The .xml file can be utilized in the later training of network. In Figure 11, the original pictures in the dataset are annotated by Labelimg software to generate the .xml files. Among them, the original image is given in Figure 11a. The image after labeling the target gesture through Labelimg software is given in Figure 11b. The .xml file generated after image annotation is given in Figure 11c. The .xml file contains the information such as the reading route of the files, the size of the pictures, the target category, the location of the target boxes, and the sizes of the target boxes.

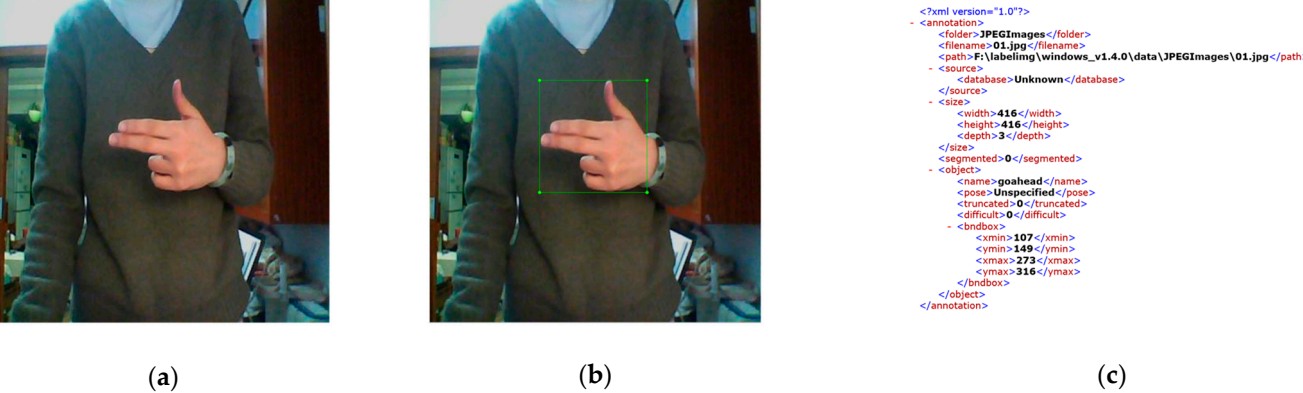

|  (a)  |  (b)  |  (c)  |

**Figure 11.** Dataset annotation map: (**a**) unlabeled picture; (**b**) annotated image; (**c**) generated .xml tag file.

## 4. Network Training and Model Effect Analysis

### 4.1. Model Performance Evaluation

The network ROC curve, PR curve, network mAP value, and the time which is taken to recognize randomly selected images are selected as the performance evaluation indicators. The YOLOv4-gesture model is combining with k-means++, the CBAM and SPP mechanism on the basis on the YOLOv4 model. The YOLOv4-motion-blur-gesture model adds the DeblurGanv2 network basis on the YOLOv4-gesture model to preprocessing the blur images. The network training and effect comparison are carried out under the same parameters. The evaluation of the model is judged from two dimensions: accuracy and real-time. The schematic diagram of network sample allocation is shown in Figure 12. In Figure 12, the positive samples that are correctly classified are represented as True Positives (TP). The negative samples that are correctly classified are defined as True Negatives (TN). The negative samples that are incorrectly assigned as positive samples are represented as False Positives (FP). The positive samples that are incorrectly assigned as negative samples are represented as False Negatives (FN). The ratio of correctly classified positive samples among all positive samples is called accuracy. The ratio of the correctly classified positive samples to all the actual positive samples is called recall rate.

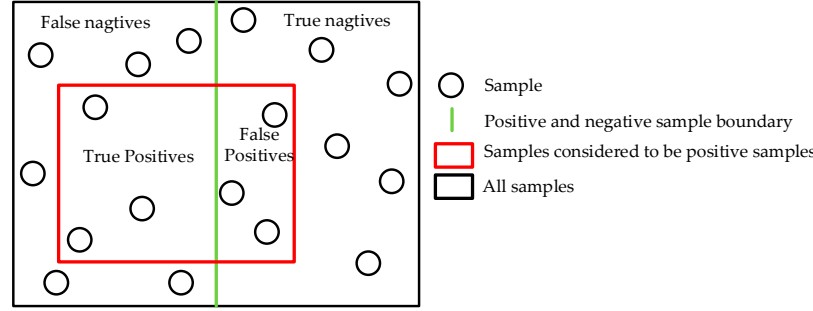

**Figure 12.** Network sample distribution diagram.

### 4.2. Effect Analysis of Motion Blur Gesture Detection

The target recognition algorithm based on the deep learning is utilized to complete the recognition of static gestures. The training of the deep learning neural network should be done first to complete the automatic recognition task of static gestures. After the corresponding algorithm is given, the features of an image in the dataset can be learned by training the neural network and generating weights at the same time. The weight out is the criterion for finding and judging the target. Then, the task of recognizing the target gesture can be completed by writing the weight file into the model algorithm. The processing of network input motion blur images is realized through the learning and training of dynamic blur and clear images through the network.

### 4.2.1. Image Motion Blur Reduction Effect

Motion blurred images exist widely in the process of human–computer interaction, which interferes with target recognition. To make the network have better detection and recognition ability for motion blurred images, DeburGanv2 network module is inserted before the input channel recognized by YOLOv4model. The Deburganv2 network module is used to process the input unknown blur kernel image to realize the optimization of the motion image. The conclusion is obtained by comparing the effect and processing time between the blurred images and the processed blurred images. The motion blurred image processing effects of four different gestures compared in the experiment are shown in Figure 13. The DeburGanv2 network has an excellent processing effect for four kinds of gestures and can remove the motion blur in the image well. The MobileNet is used as the backbone network to train DeburGanv2 network. In terms of processing time, the YOLOv4 target detection network added to the DeburGanv2 network to process blurred images has increased by 10%. In terms of network recognition effect, the accuracy of YOLOv4 target detection model with the DeburGanv2 network for blur image processing has been increased by 30%.

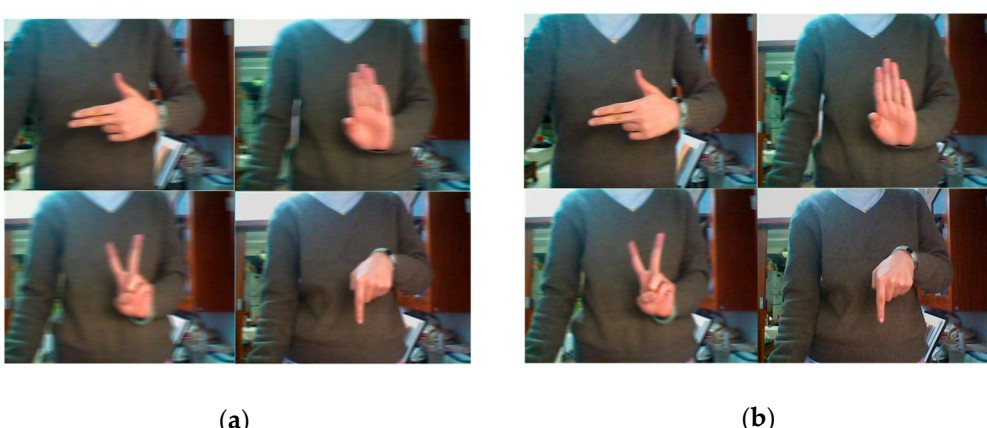

(**a**)         (**b**)

**Figure 13.** Motion blurred image processing effects of four different gestures compared in the experiment: (**a**) unprocessed motion blurred gesture images; (**b**) motion blurred image processed by the DeblurGanv2 model.

### 4.2.2. Motion Blur Gesture Detection Accuracy and Recognition Speed Effect

To obtain more reliable clustering results, the K-means++ clustering algorithm is used to replace the original K-means clustering algorithm. The prior frame sizes calculated under different clustering algorithms are used for network training. The frame selection effects of the two methods are compared to draw a conclusion. The network detection effect chart under the k-means and k-means++ clustering algorithms is shown in Figure 14. The experiment compared the frame selection effects of four different gestures. The frame selection integrity of victory gesture and the accuracy of target recognition are the most different. The improvement of the priori boxes clustering algorithm achieves good results. In Figure 14, the using of k-means++ clustering algorithm is not prone to the invalid selection. The k-means++ clustering algorithm has higher frame selection integrity for target gestures. In contrast, the more complete target gesture selection effect can be obtained by using the k-means++ clustering algorithm to reset. The training effect of the network is improved with the improvement of the superior target selection degree. The above improvements play a important role in the recognition of specific targets.

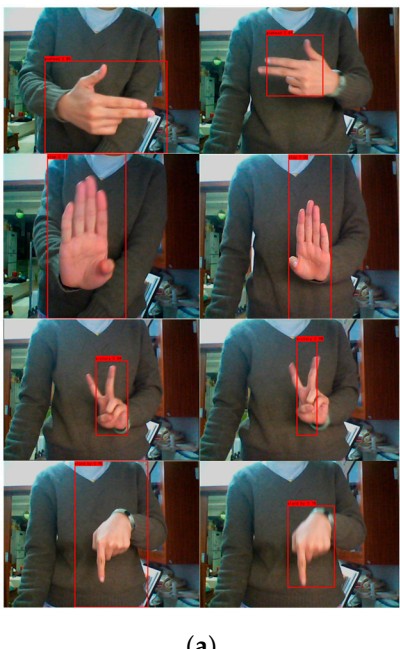
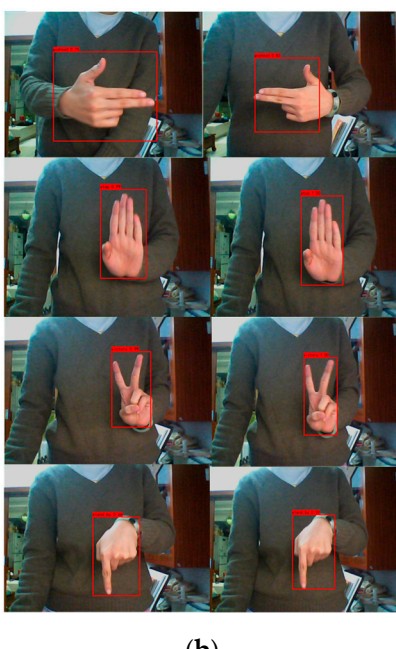

(**a**)                    (**b**)

**Figure 14.** Target frame selection effect comparison chart: (**a**) network detection effect under k-means clustering algorithm; (**b**) network detection effect under k-means++ clustering algorithm.

To verify the effectiveness of adding the SPP and CBAM module in the YOLOv4 model, the ablation experiments are carried out in this article. The SPP module and CBAM module are added into the YOLOv4 to build the YOLOv4-SPP model and YOLOv4-CBAM model. The YOLOv4-gesture model is obtained by collaborating with the YOLOv4-SPP model and YOLOv4-CBAM model. The three models are trained with the same parameters. The changes of mAP of the three models during training are shown in Table 4. Based on Table 4, it can be concluded that the addition of the SPP and CBAM modules improved the performance of YOLOv4 network. When the epoch is 100, the mAP value of the YOLOv4-SPP model improved by 2.37%, the YOLOv4-CBAM model improved by 4.96%, the YOLOv4-gesture model improved by 8.28%. Therefore, the proposed YOLOv4-gesture model is effective.

**Table 4.** The mAP values of the three models under different epochs.

| Model | mAP (epoch = 50) | mAP (epoch = 75) | mAP (epoch = 100) |
|---|---|---|---|
| YOLOv4 | 60.33% | 73.66% | 88.48% |
| YOLOv4-SPP | 64.76% | 78.21% | 90.85% |
| YOLOv4-CBAM | 64.83% | 83.74% | 93.44% |
| YOLOv4-gesture | 65.35% | 85.57% | 96.76% |

The performance effect of the network under different task requirements can be reflected by the ROC curves. In different application tasks, different cut-off points can be selected according to requirements to compare the performance of the model. The more forward the position is, the higher the precision is required. On the contrary, it requires higher recall. The ROC curves are drawn with the false positive rate of the horizontal axis and the true rate of the vertical axis. The threshold range of true positive rate is from 0 to 1. Therefore, the closer the curve is to the (0,1) point, the more comprehensive the network effect. The ROC curves of different structure networks are shown in Figure 15. The YOLOv4-motion-blur-gesture is the final network model of the experiment. To facilitate the evaluation of the comprehensive ability of the model, the straight line passing through the point (0,1) at an angle of 45° to the vertical axis is drawn in Figure 15. In Figure 15, P is the intersection point between the straight line and the ROC curve of YOLOv4-motion-

blur-gesture model. The most comprehensive effect of the YOLOv4-motion-blur-gesture model is represented by the cut-off point P. In Figure 15, the classification effect of the YOLOv4-motion-blur-gesture model is better than YOLOv4-gesture and YOLOv4 model under any limit value condition. The best effect is the YOLOv4-motion-blur-gesture model, it is closest to the (0,1) point.

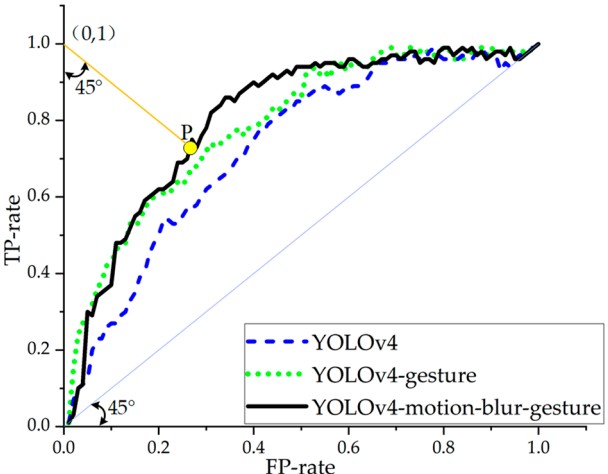

**Figure 15.** ROC curve graph of three models.

The ratio of the truly positive samples in the predicted positive results is represented as the accuracy rate (Precision). The correct predicted sample ratio in the original sample is referred to as Recall. To facilitate the evaluation of the comprehensive ability of the model, the straight line passing through the point (0,0) at an angle of 45° to the horizontal axis is drawn in Figure 16. The intersection between the PR curve and diagonal line is the classification result under the comprehensive consideration of the network. The relationship between the precision rate and the recall rate is represented by the PR curve. The network PR graph and mAP values of three models are shown in Figure 16. In Figure 16, it can be concluded that the YOLOv4-motion-blur-gesture model has the best effect. The effect of YOLOv4 is relatively poor. In target detection, the average accuracy (AP) is the area under the PR curve. The mAP value of the YOLOv4-motion-blur-gesture is higher than other two models, reaching 96.76%.

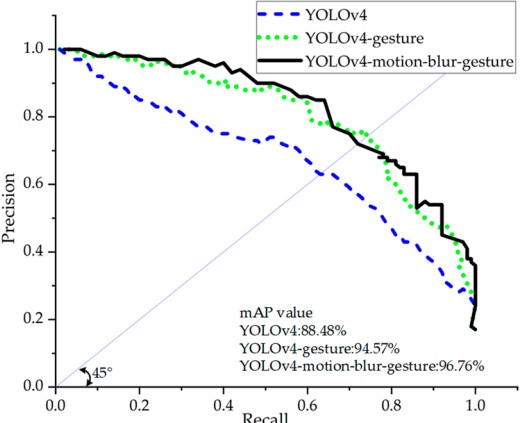

**Figure 16.** PR graph of three models.

In the real-time monitoring reference environment, the speed of network target recognition also affects the final work efficiency. The improved YOLOv4-gesture and YOLOv4-motion-blur-gesture models are used to recognize 100 randomly selected gesture pictures. The identification time sampling diagram of the three models is shown in Figure 17. The

average recognition time of YOLOv4-gesture and YOLOv4-motion-blur-gesture models is longer than the YOLOv4 model. Moreover, whether the input is a blurred image or a clear image, the performance of YOLOv4-motion-blur-gesture model is more stable. In a good interactive environment, the YOLOv4-motion-blur-gesture model may take longer time than YOLOv4 model. However, in the aerospace space with harsh environment, blurred images exist widely, the YOLOv4-motion-blur-gesture model proposed in this article has higher reliability.

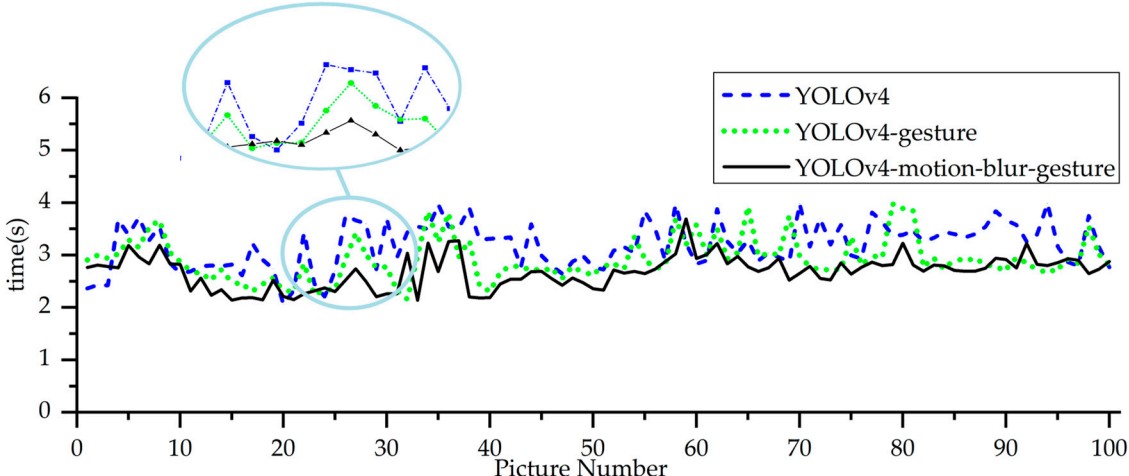

**Figure 17.** Identification time sampling diagram of three models.

It can be seen from Figures 14–17 that the YOLOv4-motion-blur-gesture model proposed in this article has made great progress in motion blurred gesture detection. The accuracy, precision, recall, mAP value of target detection, and the detection speed of motion blurred image are improved in the YOLOv4-motion-blur-gesture model. To express the improvement effect obtained by the YOLOv4-motion-blur-gesture model more intuitively, the mAP value and the average detection time of the three models are shown in Table 5. In Table 5, the average detection accuracy and the mAP value of the YOLOv4-gesture model are improved by 7.07% and 6.09%, respectively, compared with the YOLOv4 model. The YOLOv4-gesture model also reduces the average detection time of targets by 0.27 s. Compared with the YOLOv4-gesture model, the average detection accuracy and the mAP value of YOLOv4-motion-blur-gesture model are improved by 1.37% and 2.19%, respectively. The YOLOv4-motion-blur-gesture model also reduced the average detection time of the target by 0.24 s. Compared with the YOLOv4 model, the YOLOv4-motion-blur-gesture model has a greater improvement. The average detection accuracy and the mAP value were improved by 8.44% and 8.28%, respectively. The YOLOv4-motion-blur-gesture model also reduced the average detection time of the target by 0.51 s.

**Table 5.** Average detection accuracy, mAP, and average detection time of three models.

| Model | Average Detection Accuracy | mAP | Average Detection Time |
|---|---|---|---|
| YOLOv4 | 89.35% | 88.48% | 3.19s |
| YOLOv4-gesture | 96.42% | 94.57% | 2.92s |
| YOLOv4-motion-blur-gesture | 97.79% | 96.76% | 2.68s |

## 5. Conclusions

The recognition effect of network on the gestures of motion blur was researched in this article. In the process of human–computer interaction, the phenomenon of image motion blur is produced by the movement between the hand and the camera. The motion blur affects the network recognition and is not easy to avoid. To improve the speed and accuracy

of motion blurred gestures recognition, the research of YOLOv4 algorithm was studied from the two aspects of motion blurred image processing and model optimization. The DeblurGanv2 model is used to remove the motion blur of the gestures in YOLOv4 network input pictures. The K-means++ algorithm is used to cluster the priori boxes for obtaining the more appropriate size parameters of the priori boxes. The CBAM attention mechanism and SPP spatial pyramid pooling structure are added to YOLOv4 model to improving the efficiency of network learning. The dataset for network training is designed for the human–computer interaction in aerospace space. To reduce the redundant features of the captured images and enhance the effect of model training, the Wiener filter and bilateral filter are superimposed on the blurred images in the dataset to simply remove the motion blur. The augmentation of the model is executed by imitating different environments.

The network structure optimization is obtained through experimental comparison. A YOLOv4-gesture model is built, which collaborates with K-means++ algorithm, the CBAM and SPP mechanism. A DeblurGanv2 model is built to process the input images of the YOLOv4 target recognition. The YOLOv4-motion-blur-gesture model is composed of the YOLOv4-gesture and the DeblurGanv2. Based on the original data set used in model training, the augmented simulation under aerospace conditions was carried out to complete the expansion. The YOLOv4-gesture model and YOLOv4-motion-blur-gesture model were compared with the YOLOv4 model by drawing the ROC curves, PR curves, and time curves for the image detection and mAP histogram in the experiment. The proposed model has a stable recognition effect in the real-time interaction of motion blur gestures, it improves the network training speed by 30%, the target detection accuracy by 10%, and the value of mAP by about 10%. The experimental results show that the YOLOv4-motion-blur-gesture model is reasonable and can be better adapted to the real-time human–computer interaction application.

The YOLOv4-motion-blur-gesture model proposed in this article can well meet the motion blur gesture recognition in the real-time interactive environment. The model proposed in this article realizes the real-time recognition of blur images with high accuracy without much change in the detection speed. The radiation, space station turbulence, and hand motion blur always present in the aerospace environment. In the harsh space operation environment, it can realize high inclusive gesture recognition. The model also greatly improved the training speed and can carry out targeted training for specific application environment to get better results. Further research should be based on the extended data set of simulated aerospace environment to carry out the simulation experiments in special environments. The research results of this article can not only provide a technical basis for the multimodal human–computer interaction of electrically driven large-load-ratio multi-legged robot in interstellar exploration, but also can be applied to detection and interaction scenes in other complex environments with high real-time requirements.

**Author Contributions:** Conceptualization, H.Z.; methodology, H.Z. and Y.X.; formal analysis, N.W. and L.D.; data curation, H.Z. and Y.X.; writing—original draft preparation, H.Z. and Y.X.; writing—review and editing, H.Z.; visualization, Y.X. and N.W.; supervision, H.Z.; funding acquisition, H.Z. All authors have read and agreed to the published version of the manuscript.

**Funding:** This research was funded by the National Natural Science Foundation of China, grant number 51505335; the Doctor Startup Projects of TUTE, grant number KYQD 1806.

**Institutional Review Board Statement:** Not applicable.

**Informed Consent Statement:** Not applicable.

**Data Availability Statement:** Not applicable.

**Conflicts of Interest:** The authors declare no conflict of interest.

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
