# Peer review of "High Inclusiveness and Accuracy Motion Blur Real-Time Gesture Recognition Based on YOLOv4 Model Combined Attention Mechanism and DeblurGanv2"

_applsci, doi:10.3390/app11219982_

Round 1
Reviewer 1 Report
The authors have reported the study on YOLOv4 gesture detection considering attention mechanism and motion blurring. In general, the main conclusions presented in the paper are supported by the figures and supporting text. However, to meet the journal quality standards, the following major comments need to be addressed
- IoU, P_r should be defined in Eq 1.
- In page 3, the authors have mentioned YOLO is “proposal-free regression method”, please elaborate.
- Typo in page 5, line : 199 : “neural ne t works”
- Page 5: The author has mentioned “CBAM model has the advantages of simple structure, small computation, and fast computation speed”. In the current study CBAM has been added in PANet replacing CBL/CBM type block, did the author performed an ablation study to justify the claim?
- Page 6: “CBAM module is inserted among the three channels output of the CSPNet structure”..shouldn’t be CSPDDarknet 53?
- Not quite sure why authors should add SPP in two other sampling outlet of the backbone has been added (Fig.7) ? It’s a plug-in module in original YOLOv4 from the smallest scaled convolution block. Please elaborate this point ?… did author performed any ablation study to check any accuracy improvement?
- The authors should add the numbers of training data for each class, if they used any data augmentation methods, training time and if they used transfer learning during training.
- The authors should add a table proving F-1, mAP, and other accuracy parameters to compare the three detection models.
- Fig 15 PR plot: why PR characteristic is not tangentially orthogonal when R is close to 1 ? This characteristic is stronger in YOLO, which should not be the case.
Results and discussion: - The paper is overall descriptive can be improved. Comparison with existing literature may be incorporated , limitations of the results should be discussed. It would be interesting to discuss whether the model presented here can be used in other YOLO v4 applications [ see : AI 2021, 2(3), 413-428; https://doi.org/10.3390/ai2030026; Sensors 2021, 21(9), 3263;https://doi.org/10.3390/s21093263] . Hence should be addressed in the introduction
- Typographical errors: There are several minor grammatical errors and incorrect sentence structures. Please run this through a spell checker.
Reviewer 2 Report
I enjoyed reading your article on an interesting topic. In this paper, the YOLOv4-motion-blur-gesture model was proposed.
Even dark images can be regarded as a kind of noise.
In this thesis, was the experiment performed in an environment such as a dark image? Also, it seems that there will be errors in object recognition depending on the background of the hand.
I wonder if the experiment was performed in the various environments mentioned above in this experiment.
It would be good if you could mention more specifically about the experimental data and environment mentioned above
Round 2
Reviewer 1 Report
The authors address all the reviewer's queries. The manuscript can be published in its current form.